# Analysis of Phytohormone Signal Transduction in *Sophora alopecuroides* under Salt Stress

**DOI:** 10.3390/ijms22147313

**Published:** 2021-07-07

**Authors:** Youcheng Zhu, Qingyu Wang, Ziwei Gao, Ying Wang, Yajing Liu, Zhipeng Ma, Yanwen Chen, Yuchen Zhang, Fan Yan, Jingwen Li

**Affiliations:** College of Plant Science, Jilin University, Xi’an Road, Changchun 130062, China; yczhu19@mails.jlu.edu.cn (Y.Z.); qywang@jlu.edu.cn (Q.W.); gaozw8218@mails.jlu.edu.cn (Z.G.); wangying2009@jlu.edu.cn (Y.W.); yj_liu@jlu.edu.cn (Y.L.); mazp20@mails.jlu.edu.cn (Z.M.); chenyw8218@mails.jlu.edu.cn (Y.C.); yczhang19@mails.jlu.edu.cn (Y.Z.)

**Keywords:** *Sophora alopecuroides*, phytohormone signal transduction pathways, salt stress, differentially expressed genes, differential metabolites

## Abstract

Salt stress seriously restricts crop yield and quality, leading to an urgent need to understand its effects on plants and the mechanism of plant responses. Although phytohormones are crucial for plant responses to salt stress, the role of phytohormone signal transduction in the salt stress responses of stress-resistant species such as *Sophora alopecuroides* has not been reported. Herein, we combined transcriptome and metabolome analyses to evaluate expression changes of key genes and metabolites associated with plant hormone signal transduction in *S. alopecuroides* roots under salt stress for 0 h to 72 h. Auxin, cytokinin, brassinosteroid, and gibberellin signals were predominantly involved in regulating *S. alopecuroides* growth and recovery under salt stress. Ethylene and jasmonic acid signals may negatively regulate the response of *S. alopecuroides* to salt stress. Abscisic acid and salicylic acid are significantly upregulated under salt stress, and their signals may positively regulate the plant response to salt stress. Additionally, salicylic acid (SA) might regulate the balance between plant growth and resistance by preventing reduction in growth-promoting hormones and maintaining high levels of abscisic acid (ABA). This study provides insight into the mechanism of salt stress response in *S. alopecuroides* and the corresponding role of plant hormones, which is beneficial for crop resistance breeding.

## 1. Introduction

Salt stress severely restricts the ability to improve crop yield and quality, which is an issue with increasing impact owing to global changes in the climate and environment [1]. To address this problem, it is important to improve the salt tolerance of crops [2]. One promising factor is that plants have developed a series of abilities to resist salt stress during long-term evolution [3]. Differences in the environments of plants lead to differences in salt tolerance [3,4]. To successfully cultivate highly salt-tolerant crops, it is necessary to further explore the salt tolerance of highly resistant plants [4]. *Sophora alopecuroides* is a legume plant that can adapt to harsh natural environments and exhibits strong stress resistance [4,5]. Currently, there are few studies on the effects of salt stress on *S. alopecuroides* and on the mechanism of its response to salt stress.

Different plants have various strategies for responding to salt stress with the purpose of reducing the impact of the stress. This may be accomplished by enhancing stress tolerance or by avoiding the salt through reduction of salt ion concentrations [1,2]. Plants enhance stress tolerance through a series of physiological and biochemical reactions, including the perception of stress signals, signal transduction, transcription, and metabolic responses [3]. Phytohormones are small chemicals that play key roles in plant growth and development [6]. Studies have shown that phytohormones also play important roles in the molecular signaling of plants in response to environmental stress [6]. Plant hormones include auxin (AUX), cytokinins (CKs), gibberellin (GA), ethylene (ETH), abscisic acid (ABA), salicylic acid (SA), jasmonic acid (JA), brassinosteroids (BRs), and strigolactones (SLs) [7]. The biological functions of plant hormones are not singular and they may play different roles in different plants, tissues, and growth stages and under different environmental conditions [6,8,9,10,11]. Plant hormones are classified according to their function in plant growth and stress adversity, with AUX, CKs, GA, BRs, and SLs being classified as growth-promoting hormones and ABA, SA, and JA regarded as stress response hormones [7]. AUX plays important roles in biological processes such as apical dominance, embryonic development, adventitious root formation of lateral roots, and differentiation of vascular tissues [12]. AUX is sensed by receptors and forms SKP1, Cullin, and F-box (SCF) complexes, which binds to AUX/IAA inhibitors and is involved in ubiquitination and proteasome-mediated degradation of AUX/IAA, the release AUX response factors (ARF), and activation of AUX-induced gene expression [13]. *Arabidopsis* AUX receptor mutants are more sensitive to salt stress and the AUX receptor genes *TIR1* and *AFB2* are downregulated under salt stress, which indicates that *Arabidopsis* slows plant growth to improve salt tolerance by maintaining a low AUX signal response [14,15]. Meanwhile, CKs are involved in cell division, reproductive development, leaf senescence, regulation of root-shoot ratios, and adaptation to abiotic stress during plant growth and development [16,17]. CKs are sensed by receptors AHK2/3/4 located on the cell membrane and activate B-type transcription factor ARRs through phosphorylation [18]. A CK receptor AHK2/3/4 mutant showed stronger tolerance to salt stress and the downstream gene *AHP2/3/5* and mutations in B-type response modifiers can improve salt tolerance of plants [11,19]. CK is also considered a communication messenger between the roots and aboveground parts of plants during salt stress [20]. The decrease in CK levels and increase in ABA synthesis in plants under salt stress are considered effective defense mechanisms for plants responding to salt stress [6]. In comparison, BRs regulate plant salt tolerance by interacting with other signaling molecules, inducing the production of ETH and hydrogen peroxide and activating antioxidant enzyme activity [21,22]. It has been reported that GA plays a role in promoting stem elongation, regulating the development of meristems, and regulating biotic and abiotic stresses [23,24]. GA binds to the receptor GOD1, induces the conformation of GOD1 to change, and then binds to the DELLA protein to form a GA-GID1-DELLA complex, which leads to degradation of the DELLA protein by the 26S proteasome and the activation of downstream response genes [25]. Reduction of GA levels causes a slowing in plant growth and helps improve stress resistance [26]. Meanwhile, ETH is a small-molecule gas plant hormone that is widely used in agriculture [27,28]. ETH promotes flowering, seed germination, leaf senescence, fruit ripening, and other physiological functions and biochemical reactions [27,29]. ETH accumulates in plants under salt stress and *Arabidopsis thaliana* treated with ACC shows enhanced salt tolerance at different growth and development stages [30,31,32]. The JA biosynthesis mutant caused by a mutation in allene oxide synthase has a lower ABA content, whereas an ABA biosynthesis mutant has a lower JA content [33]. The JA–ABA interaction plays an important role in salt responses of plants [6]. ABA is mainly synthesized in vascular tissues and then transported to guard cells to respond to osmotic stress and salt stress by regulating stomata [34]. As the main mediator of plant responses to stress, ABA can improve plant survival under salt stress by activating plasma membrane binding channels or by combining with Ca^2+^ [35]. The main pathway of SA biosynthesis primarily occurs in the chloroplast with the SA being conjugated in the cytoplasm and stored in vacuoles [36]. SA in wheat prevents the decline in levels of CK and AUX to promote growth, while also maintaining high ABA levels to enhance salt tolerance [37].

Current research on plant hormones with respect to plant salt tolerance has mainly focused on the model species of *A. thaliana*, rice, corn, and tomato [38,39,40,41]. There are few studies on the mechanism of phytohormones in salt tolerance of species with strong stress resistance, and there are no reports on phytohormones in salt-stressed *S. alopecuroides*. In the present study, transcriptome analysis methods were used to evaluate salt-stressed *S. alopecuroides* at the transcriptional level at different time periods. This was combined with non-targeted metabolome analysis to detect changes in levels of *S. alopecuroides* metabolites under salt stress. Short time-series expression miner (STEM) was used to analyze the expression trends of differentially expressed genes (DEGs) and differential metabolites (DMs) affected by salt stress, combined with transcriptome and metabolome analyses to evaluate the role of plant hormone signal transduction pathways in response to salt stress. This study provides a reference point for further exploration of the salt tolerance genes of *S. alopecuroides* and to exploit their use, as well as guide their significance in the study of plant hormones and plant salt tolerance.

## 2. Results

### 2.1. Transcriptome Analysis and STEM Analysis of DEGs

To explore the effect of salt stress on *S. alopecuroides* and its response, we treated *S. alopecuroides* under hydroponic conditions to ensure a single stress factor (Figure 1A). A video demonstrating the phenotypic changes of *S. alopecuroides* caused by salt stress during the sampling process and hydroponic treatment is provided in Appendix A. In the current study, transcriptome sequencing analysis was performed on *S. alopecuroides* roots, and 342,516,112 raw sequences and 331,205,836 clean reads were obtained from the control and treatment groups. Transcripts and UniGenes 501–2000 bp in length accounted for 53.08% and 51.37% of their respective populations (Figure 1B). Expression trend analysis of all DEGs revealed six significant change trends (Figure 1C–H), of which three were upregulated. The trend of upregulated expression of DEGs primarily occurred at 4 h, 24 h, and 72 h after induction of salt stress. These results were consistent with the phenotypic trends observed in salt-treated *S. alopecuroides*.

### 2.2. STEM Analysis of DMs

To investigate the changes in the levels of *S. alopecuroides* root metabolites after salt stress and the possible benefit they confer to the responses to salt stress, we analyzed the changes in DMs. The results revealed there were eight significant change trends (Figure 2), four of which were upregulated with metabolites gradually accumulating during salt stress, and four were downregulated. Salt stress, therefore, induces a strong stress response in the roots of *S. alopecuroides*. The upregulated metabolites may have included protective substances as well as harmful substances that formed and accumulated in response to salt stress. Identification of the changes in levels of these metabolites helped us further analyze the mechanism of *S. alopecuroides* in response to salt stress.

### 2.3. DEGs Were Significantly Enriched in Plant Hormone Signal Transduction

The DEGs identified were quantified under each expression trend. The DEGs whose expression trends were mainly upregulated or downregulated were re-annotated in Kyoto Encyclopedia of Genes and Genomes (KEGG) metabolic pathway maps (Figure 3). The results showed the DEGs were mainly annotated in the plant hormone signal pathway, indicating that plant hormone signal transduction plays an important role in the response of *S. alopecuroides* roots to salt stress. We further analyzed the DEGs annotated in the signal transduction and biosynthetic pathways of plant hormones and combined this with the changes in DMs to better delineate the role of plant hormones in the salt stress response in *S. alopecuroides*.

### 2.4. AUX, CKs, GA, and BRs Regulated S. alopecuroides Growth under Salt Stress

Further analysis revealed that DEGs in the AUX, CKs, GA, and BR signaling pathways were significantly downregulated at 4 h and 72 h after initiation of salt stress and there were no significant (*p* < 0.05) changes in subsequent expression levels from 24 h to 48 h. Phenotypic observation of the plants showed the growth state of *S. alopecuroides* was normal from 24 h to 48 h post salt stress, indicating these four growth-promoting hormones may have played a role in promoting growth recovery in response to salt stress. 

We identified four core response genes in the AUX signal transduction pathway of *S. alopecuroides* that were significantly downregulated at 4 h and 72 h under salt stress, *SaARF-1*, *SaARF-2*, *SaARF-3*, and *SaARF-4*. However, expression was restored at 24 h and 48 h under salt stress, which indicated *S. alopecuroides* may have resumed growth at this stage. The expression trends for *SaGH3*, *SaIAA*, and *SaSAUR*, which are downstream genes regulated by ARF, were similar to those of *SaIAA1*, *SaIAA2*, *SaIAA3*, *SaIAA4*, *SaIAA5*, *SaGH3-1*, *SaSAUR4*, *SaSAUR5*, *SaSAUR6*, and *SaSAUR7*, consistent with the changing trend observed for ARF, whereas those for *SaIAA14*, *SaIAA15*, *SaIAA16*, *SaIAA17*, *SaIAA18*, *SaGH3-4*, *SaSAUR9*, *SaSAUR10*, and *SaSAUR11* were completely opposite to that of ARF. Therefore, the roots of *S. alopecuroides* maintain growth regulation during the response to salt stress (Figure 4). Further analysis of DEG and DM trends identified in the AUX biosynthetic pathway revealed that levels of the AUX precursor indole increased at 24 h under salt stress. The tryptophan and tryptamine contents were also upregulated. After 24 h under salt stress, *S. alopecuroides* roots may, therefore, have maintained their growth by increasing AUX levels.

In the CK signal transduction pathway, 15 cytokinin response 1 (*CRE1*) genes that serve as CK receptors were determined to be differentially expressed after salt stress, of which nine were significantly upregulated and six were significantly downregulated. One *SaAHP* gene was also differentially expressed, with its expression level being highest at 4 h post salt stress induction and gradually decreasing from 24 h to 72 h. One type-B ARR and three type-A ARRs were identified as being differentially expressed. Among them, *SaARR-B* was upregulated in the early stage of salt stress, but downregulated at 72 h. Changes in *SaARR-A-1* and *SaARR-A-2* expression were consistent with that of *SaARR-B*; however, *SaARR-A-3* expression was significantly downregulated after salt stress. These results led us to speculate the CK-mediated signaling pathway positively regulated the growth of *S. alopecuroides* during the initial stage of salt stress. We further explored change trends in the upstream DEGs and their DMs of CK after salt stress in the roots of *S. alopecuroides* (Figure 5). Adenosine 5’-monophosphate (AMP) was significantly upregulated after exposure to salt stress, indicating that the roots of *S. alopecuroides* may have increased energy production in response to salt stress in order to maintain growth and development. The CK biosynthesis-related genes *IPT* and *CYP735A* were significantly downregulated at 4 h after salt stress, upregulated at 24 h and 48 h, and then downregulated again at 72 h. These results were consistent with the phenotypic changes observed for *S. alopecuroides* after salt stress and indicated that CK production by *S. alopecuroides* was directly related to its growth state.

The expression of DELLA, a key inhibitor of the GA signaling pathway, was significantly downregulated after salt stress. Its expression level was lowest at 72 h after salt stress, but had not changed significantly at 4 h, 24 h, and 48 h. No negatively regulated downstream DEGs were identified. These results indicate the downregulated expression of *DELLA* may have been caused by decreased GA levels after salt stress. Both *ECDS* and *GA20ox*, key genes for GA biosynthesis, were downregulated and the expression of the GA hydrolysis gene *GA2ox* was significantly upregulated at 4 h (Figure 6). GA signal transduction, therefore, plays an important role in the response of *S. alopecuroides* roots to salt stress.

No BR-related metabolites were found, which may have been because of the relatively low content levels of BR and its metabolites in root tissue of *S. alopecuroides*. However, we identified multiple DEGs in the BR biosynthesis and BR signal transduction pathways. Among them, *SaCYP90B1* and *SaCYP85A1*, key genes involved in BR biosynthesis, were significantly upregulated at 4 h and 24 h post induction of salt stress. However, *SaCYP90B1* was downregulated at 72 h, while *SaCYP85A1* was upregulated. We found the expression of four *CYCD3* genes regulated by BR signaling was downregulated at 4 h and 24 h under salt stress, their expression restored at 48 h, and the expression levels were the lowest at 72 h (Figure 6). Four core regulatory genes of BR signaling were identified, *SaBSK-1*, *SaBSK-2*, *SaBSK-3*, and *SaBSK-4*. Among them, *SaBSK-1* and *SaBSK-2* were upregulated by salt stress, while *SaBSK-3* and *SaBSK-4* were downregulated. This indicates that BR signaling may have played a role in regulating *S. alopecuroides* growth under salt stress.

### 2.5. JA and ETH Signals Negatively Regulate S. alopecuroides in Response to Salt Stress

Expression of negative regulators of the JA and ETH signaling pathways in the root tissues of *S. alopecuroides* was significantly upregulated under salt stress. Specifically, *JAZ*, a negative regulatory of the JA signal pathway, was upregulated, with its expression being highest at 24 h under salt stress (Figure 7). Furthermore, we found that JA and MeJA were significantly reduced under salt stress. The JA receptor gene was also downregulated. Collectively, this indicates that JA may have been a negative regulatory in *S. alopecuroides* roots in response to salt stress.

The levels of adenosylmethionine of the ETH biosynthesis pathway were significantly reduced at 48 h and 72 h under salt stress. The annotation results for DEGs of the ETH signal transduction pathway showed that *ETR*, a negative regulator of ETH signaling, was significantly upregulated at 4 h and 24 h under salt stress, decreased at 48 h relative to that at 24 h, and increased again at 72 h (Figure 7). This indicates that ETH signaling may play a negative regulatory role in the response of *S. alopecuroides* roots to salt stress.

### 2.6. ABA and SA Levels Increased to Enhance S. alopecuroides Tolerance to Salt Stress

Observation of *S. alopecuroides* phenotypic changes under salt stress revealed mild wilting in the first 4 h after salt stress. The plants recovered from this phenotype after 4 h, with the change not being obvious at 24 h or 48 h. However, the leaves withered, with the old leaves falling off by 72 h. We found that ABA levels were significantly upregulated at 24 h post salt stress and SA levels were consistently upregulated at all times after salt stress. ABA and SA may, therefore, play positive regulatory roles in the response of *S. alopecuroides* roots to salt stress; increased levels of these plant hormones may improve *S. alopecuroides* resistance. We found that three core genes of ABA signal transduction, *SaABF-1*, *SaABF-2*, and *SaABF4*, were upregulated under salt stress and their expression levels were highest at 4 h and 24 h (Figure 8). Therefore, the ABA signal transduction pathway participates actively in the response of *S. alopecuroides* roots to salt stress. In addition, expression of the ABA receptor genes *SaPYL*, *SaPYL1*, and *SaPYL4* was significantly downregulated at 4 h under salt stress, while the expression of *PP2C* was highest at 4 h and 24 h (Figure 8). Additional analysis of ABA biosynthesis-related genes revealed that their expression levels were increased at 4 h and 24 h, further indicating that ABA positively regulated *S. alopecuroides* resistance to salt stress.

The levels of SA in the roots of *S. alopecuroides* gradually increased under salt stress, as did the levels of trans-cinnamic acid, l-phenylalanine, d-phenylalanine, and *N*-acetyl-l-phenylalanine. This suggests that SA acts as a positive regulator in the response to salt stress. By analyzing the DEGs of the SA signal transduction pathway, two NPR1 genes were identified, *SaNPR1-1* and *SaNPR1-2*, in addition to one *SaTGA* gene and five SaPR-1 genes. We found that *SaPR-1* was significantly upregulated under salt stress. Meanwhile, *SaTGA* was initially upregulated under salt stress, but then downregulated, reaching its highest level at 4 h (Figure 8). The expression of *SaPAL*, a key gene in SA biosynthesis, was upregulated at 4 h, 24 h, and 72 h under salt stress, but there was no significant (*p* < 0.05) difference in its expression at 48 h compared with that of the control [4]. However, *S. alopecuroides* growth was stable for 48 h under salt stress, indicating that SA played a positive regulatory role in the response of *S. alopecuroides* to salt stress.

### 2.7. Phytohormones Regulated the Balance of S. alopecuroides Growth and Tolerance under Salt Stress

To investigate the crosstalk between various plant hormones in *S. alopecuroides* roots in response to salt stress, we analyzed the trends of DEGs and DMs in each of the plant hormone biosynthesis pathways (Figure 9). This included carbohydrate metabolism (ko00020: TCA cycle; ko00030: pentose phosphate pathway), amino acid metabolism (ko00260: glycine, serine, and threonine metabolism; ko00270: cysteine and methionine metabolism; ko00360: phenylalanine metabolism; ko00380: tryptophan metabolism; ko00400: phenylalanine, tyrosine, and tryptophan metabolism), lipid metabolism (ko00592: alpha-linolenic acid metabolism), terpenoids and polyketides metabolism (ko00900: terpenoid backbone biosynthesis; ko00902: monoterpenoid biosynthesis; ko00904: diterpenoid biosynthesis; ko00906: carotenoid biosynthesis; ko00908: zeatin biosynthesis), and energy metabolism (ko00710: carbon fixation in photosynthetic organisms). The results revealed the key genes *SaCrtZ*, *SaZEP*, and *SaNCED* in the ABA biosynthetic pathway were significantly upregulated under salt stress (Appendix A). Correspondingly, downstream genes of 2E, 6E-famesylpyrophosphate, which are key genes in the BR biosynthetic pathway, were downregulated under salt stress. This shows that ABA content was increased and BR content was reduced in the roots of *S. alopecuroides* under salt stress in order to slow its growth and improve resistance. There were both upregulated and downregulated DEGs of sugar metabolism, amino acid metabolism, and lipid metabolism. This also indicates that the roots of *S. alopecuroides* actively regulate the balance between growth and resistance under salt stress.

To further verify the role of plant hormone signal transduction pathways in the response of *S. alopecuroides* roots to salt stress, we performed quantitative reverse transcription polymerase chain reaction (qRT-PCR) on randomly selected DEGs, including *SaAUX1*, *SaARF*, *SaAHP*, *SaARR-B*, *SaCYP90B1*, *SaCYP85A1*, *SaCYCD3*, *SaGA2ox*, *SaGID1*, *SaGID2*, *SaDELLA*, *SaETR*, *SaEBF*, *SaJAR1*, *SaJAZ*, *SaPYL*, *SaPP2C*, *SaABF*, *SaPAL*, and *SaTGA* (Figure 10). The results confirmed the expression levels of the 19 DEGs were consistent with the transcriptome analysis results, verifying the results were reliable. To explore protein interactions of the selected DEGs, we performed protein–protein interaction (PPI) network analysis. The analysis results predicted multiple proteins that interacted with the PAL protein of SA biosynthesis (Appendix A). This provides a reference for further study regarding the mechanism of key genes involved in signal transduction pathways of plant hormones.

## 3. Discussion

The stress on plants during the early stage of salt stress is primarily osmotic stress, while the salt stress induced by Na^+^ is more obvious during the later stages [42]. This is consistent with the trends we observed in phenotypic changes of *Sophora alopecuroides* under salt stress. From 0 to 4 h of salt stress, *S. alopecuroides* exhibited obvious water loss, but recovered beyond 4 h, indicating the regulation of osmotic stress in *S. alopecuroides* during the early stage of salt stress was completed in a short amount of time. In addition, previous studies have shown that salt-specific signals are rapidly induced in plant roots during the early stages of salt stress [43,44]. The roots of *S. alopecuroides* showed obvious changes in transcription levels in the early stage of salt stress, which was confirmed by the transcriptome results. The response of plants to salt stress is complex, but effective, and involves gene expression, changes in transcription levels, post-translational regulatory changes, and changes in protein and metabolite levels, which ultimately present as phenotypic changes [42,45]. To explore the influence of salt stress on plants, different methods have been used, such as measuring physiological indicators, ion accumulation, biological yield, and survival rates [42,46]. Tolerance of different crops to salt stress varies, indicating there may be different mechanism of action in response to salt stress [9,10]. Previous studies have shown that *S. alopecuroides* is able to maintain growth under high-salt stress, which suggests it has a high level of resistance [4,5]. In the current study, we further analyzed the role of phytohormone signal transduction pathways in the roots of *S. alopecuroides* under salt stress, which is of great significance in elucidating the mechanism involved in the response of *S. alopecuroides* to salt stress.

Plants can recover their growth abilities under conditions of salt stress, with the effects of growth-promoting hormones being indispensable. Plant hormones play vital roles, acting mainly through signal transduction pathways [1,6]. Studies have shown that mutations in the *Arabidopsis* AUX receptor genes *TIR1/AFB2/AFB3* can cause the root meristem to be more sensitive to NaCl [14]. Under mild salt stress, the development of *Arabidopsis* lateral roots is induced, with AUX playing an active positive regulatory role [47]. However, root growth under high salinity is inhibited and the accumulation of AUX is reduced [48], indicating that plant growth under salt stress is regulated by AUX. We found in the current study that expression of *SaIAA*, a negative regulator of AUX signaling, did not change significantly at 24 h or 48 h under salt stress and the AUX signal-responsive gene *SaARF* was upregulated at 48 h relative to that at 24 h under salt stress. *SaUGT74B1*, a key gene of AUX biosynthesis, had the highest expression level after 24 h of salt stress, indicating that the roots of *S. alopecuroides* may have been in a stage of growth recovery between 24 h and 48 h. AUX may have promoted *S. alopecuroides* root growth under salt stress.

CKX is a key enzyme for CK degradation and its overexpression leads to decreases in endogenous CK levels, which can enhance *Arabidopsis* tolerance to salt stress and drought stress [49]. In contrast, overexpression of *IPT8*, a key gene for CK biosynthesis, reduces *Arabidopsis* salt tolerance [50]. We found that the expression of *SaIPT* and *SaCYP735A*, key genes for CK biosynthesis, was downregulated in *S. alopecuroides* roots under salt stress, suggesting that CK content is reduced in the roots to slow growth for adaptation to salt stress. However, we also found that some CK receptor genes, such as *SaCRE1*, were upregulated after 24 h of salt stress, indicating that *S. alopecuroides* roots can restore growth by enhancing the CK signals.

Spraying plants with exogenous BR can enhance growth under salt stress [21,51,52,53]. Furthermore, BR biosynthesis gene mutants are more sensitive to salt stress and exogenous BR can alleviate the sensitivity of these mutants [22,51]. Overexpression of the BR biosynthesis gene *SoCYP85A1* in spinach enhances its tolerance to high salt stress [54]. Our current findings on the roots of *S. alopecuroides* were consistent with the findings that expression levels of BR biosynthesis genes were elevated after 4 h and 24 h of salt stress. However, the expression levels of *SaBSK* and *SaCYCD3* of the BR signal transduction pathway were not significantly different from those of the control after 24 h and 48 h of salt stress, indicating that BR participates in the regulation of salt stress tolerance of *S. alopecuroides* roots during the growth recovery stage.

Overexpression of GA metabolism-related genes, such as *OsMYB91* [55], *OsGA2ox5* [56], and *AtGA2ox7* [57], can promote GA degradation, reduce GA accumulation, slow growth, increase soluble sugar and chlorophyll content, and improve salt tolerance [58]. We found that GA3 content decreases with salt stress, GA biosynthesis genes are downregulated, and GA degradation genes are upregulated in a study of salt-stressed *S. alopecuroides*. However, the GA receptor gene *SaGID1* was significantly upregulated at 4 h and 24 h under salt stress and expression of the negative regulator of GA signal transduction *SaDELLA* was significantly downregulated at 24 h. This indicates the decrease in GA content at the initial stage of salt stress may have slowed growth to improve resistance of *S. alopecuroides* and the upregulation of GA signaling at 24 h under salt stress may have helped promote growth. In summary, sorrel beans are able to maintain growth under salt stress with AUX, CK, BR, and GA playing vital roles.

The response of ETH to salt stress varies greatly among different plants [6]. For instance, ETH signal transduction has been confirmed to promote salt tolerance in *Arabidopsis*, but in rice, ETH signals negatively regulate salt tolerance [59,60]. In *Arabidopsis*, ETR1, EIN4, and ETH, which are negative regulators of *CTR1* mutations, enhance salt tolerance [31,32,61,62,63], while mutants of the ETH-positive regulators EIN2 and EIN3/EIL1 are more sensitive to salt [32,64]. Meanwhile, mutations in positive regulator genes *MHZ6/OsEIL1* and *OsEIL2* of the ETH signaling pathway in rice improve salt tolerance and overexpression lines are more sensitive [59,60]. We found in the current study that expression of the key negative regulatory gene in the ETH signaling pathway in *S. alopecuroides* root, *SaETR*, was significantly upregulated by salt stress. This indicates that the ETH signaling pathway in the roots of *S. alopecuroides* may play a negative regulatory role. We also found that changes in the expression of DEGs in the JA signaling pathway in the roots of *S. alopecuroides* were consistent with those in the ETH signaling pathway. JA has been reported to be induced by salt stress [65]. After JA is sensed by the receptor COI1, it forms an SCF^COI1^-E3 ligase complex with SKP1 and CULLIN1, which then mediates JAZ degradation by the 26S proteasome and releases the inhibition of JA response genes (such as *MYC*), thereby activating JA signaling [65]. Studies have shown that increased JA biosynthesis in *Arabidopsis* and wheat enhances salt tolerance [66]. Tomatoes with *res* mutants exhibit high JA accumulation and are more salt tolerant [67]. In rice with the JA biosynthetic mutants *cpm2* and *hebiba*, shoots are less sensitive to salt, but no differences are found in the roots [68]. In maize, the JA biosynthetic mutant shoots and roots exhibit completely opposite responses to salt stress [69], which indicates JA signaling may have tissue specificity in response to salt stress [6]. In the current study, we found that a negative regulator of JA signaling in the roots of *S. alopecuroides* was active under salt stress. Accordingly, we believe that JA may have played a negative regulatory role in the roots of *S. alopecuroides*. Further experiments using various tissues and stages are needed to fully explore the specific regulatory effects of ETH and JA signals in salt-stressed *S. alopecuroides*.

Under salt stress and osmotic stress, endogenous ABA levels in plants increase rapidly. The increased ABA is sensed by the ABA receptors PYL/PYR, binds to protein phosphatase PP2C, and then releases the PP2C inhibition of *SnRK2s*, thereby activating SnRK2s expression [70]. SnRK2.2/2.3/2.6 phosphorylates various ABA response element (ABRE)-binding proteins (AREBs)/ABRE binding factors (ABFs), which can regulate stomatal closure and leaf senescence [70,71]. Salt stress is accompanied by higher osmotic stress and ABA-regulated stomatal closure is particularly important in the response of plants to salt stress [2,72]. In *Arabidopsis*, the potassium channel KAT1 is activated by ABA-SnRK2.6 and the K^+^ transporter 1 (AKT1) is activated by the Ca2^+^-CBL1/9-CIPK23 signal, which synergistically promotes K^+^ influx and helps improve *Arabidopsis* tolerance to salt stress [1]. In the current study, we found that key genes in the ABA signaling pathway of *S. alopecuroides* roots were significantly upregulated under salt stress (Figure 8); ABA levels were also increased. These results indicate that ABA played a positive regulatory role in the response of *S. alopecuroides* roots to salt stress. 

Exogenous SA is known to be able to promote photosynthesis in plants under salt stress [73]. SA is believed to improve the salt stress tolerance of plants and promote plant growth under salt stress [6]. The SA biosynthesis mutant *sid2* is more sensitive to NaCl [74] and the SA receptor mutant *npr1-5* shows a hypersensitive phenotype to salt [75]. However, *AtNPR1* is overexpressed in rice and very high endogenous levels of SA accumulate, which makes the rice extremely sensitive to salt and drought [76,77]. This also shows that the salt tolerance of plants is dose-dependent on SA. We found that SA levels were significantly upregulated under salt stress, indicating that SA may have a positive regulatory effect on *S. alopecuroides* roots in response to salt stress.

The growth and resistance of plants under salt stress are similar to that of a seesaw, with vigorous growth and weakened resistance, enhanced resistance, and weakened growth. The growth and resistance of *S. alopecuroides* under salt stress also conformed to this model, with the balance between resistance and growth adjusting in response to salt stress. The mechanism of action of plant hormones in response to salt stress is complicated and the crosstalk between them cannot be ignored. Mild salt stress induces a small amount of ABA and activates the AUX signal, which can induce the formation of lateral roots [47]. This causes excessive accumulation of ABA, disturbs the distribution of AUX, and inhibits the development of lateral roots [78]. Studies in tomato plants have shown that increased ABA levels under salt stress cause a significant decrease in CK levels [79,80]. ABA inhibits the expression of the key CK biosynthesis gene *IPT* through MYB2, reduces the level of CK, increases the sensitivity of plants to ABA, inhibits growth and development, and improves the adaptability of plants to salt stress [12,81,82,83,84]. Under stress, the positive regulator of the CK signaling pathway in *Arabidopsis*, ARR1/10/12 (B-ARR), can interact with SnRK2s to inhibit ABA signal transduction, while SnRK2s can phosphorylate ARR5 (A-ARR) to inhibit CK signaling [85]. Mutants of BR signal-responsive genes *BSK5* and *BIN2* in *Arabidopsis* are sensitive to ABA [86,87] and overexpression of ZmBES1/BZR1-5 in maize reduces the sensitivity to ABA [88]. Seed germination of saline-alkali land plants is dependent on the ratio of GA to ABA [89]. The negative regulator of ABA signaling, *ABI4*, can regulate transcription of the GA catabolism gene *GA2ox7* and the ABA biosynthesis gene *NCED6* [90]. In the early stage of salt stress, AUX, CK, BR, and GA levels were reduced in the roots of *S. alopecuroides* and ABA levels increased, while the corresponding growth-promoting hormone signal was weakened, and the ABA signal was significantly enhanced. This indicates that *S. alopecuroides* could slow its growth by lowering the level of growth-promoting hormone and increasing the level of ABA, which improved resistance by increasing the initial adaptability to salt stress. The ABA signal genes were downregulated at 24 h and 48 h under salt stress, while the growth-promoting hormone signal genes were upregulated, indicating that *S. alopecuroides* regained growth at this stage through self-regulation, which was consistent with the conclusions obtained from our previous research [4].

The JA signal response transcription factor MYC2 is able to activate the expression of ABA-inducible genes *AtADH1* and *RD22* to adapt to osmotic stress [91], while ABA can induce the expression of *PnJAZ1* under salt stress to slow excessive ABA signals and improve tolerance [92]. The ETH signal response gene *EIN2* can improve salt tolerance by regulating ABA biosynthesis and the expression of ABA-dependent RD29B [64], indicating that JA and ETH regulate salt tolerance of plants through crosstalk with ABA [6]. We found that the effect of ETH and JA signal transduction in the roots of *S. alopecuroides* was opposite to that of ABA signal transduction, indicating there may be mutual inhibition among ETH, JA, and ABA in *S. alopecuroides* roots. It may also be that ETH and JA slow the excessive response of *S. alopecuroides* to salt stress by reducing the accumulation of ABA, thereby contributing to *S. alopecuroides* growth under salt stress. Treatment of plants with SA under conditions of salt stress leads to the downregulation of ABA biosynthesis genes and upregulation of GA biosynthesis genes, which helps maintain plant growth under salt stress [93,94]. *S. alopecuroides* is able to grow normally under salt stress and the regulatory effects of SA may be indispensable for this ability. We suspect that SA may prevent the reduction in growth-promoting hormones, such as AUX and CK, while maintaining increased levels of ABA to improve the resistance of *S. alopecuroides* to salt stress. 

In summary, the roots of *S. alopecuroides* were predominantly subjected to osmotic stress during the initial stage of salt stress, which caused a decrease in the levels of the growth-promoting hormones AUX, CK, BR, and GA; an increase in levels of the stress hormones ABA and SA; and a decrease in ETH and JA content. These changes indicate that *S. alopecuroides* may have slowed growth to improve salt tolerance. After 24 h and 48 h of salt stress, the levels of the growth-promoting hormones increased, the levels of ABA decreased, and the growth of *S. alopecuroides* was restored. The complex crosstalk between various plant hormones may have helped maintain the balance between growth and resistance of *S. alopecuroides* under salt stress. The next experimental goal is to explore the mechanism of each hormone in response to salt stress and the specific crosstalk between various plant hormones. This will provide new insight, promote additional ideas, and serve as a reference for plant resistance breeding.

## 4. Material and Methods

### 4.1. Growth of Plants and Salt Stress

Full-grained *S. alopecuroides* seeds collected from the Korla region of Xinjiang, China were treated with 98% H_2_SO_4_ and sown in potted soil (vermiculite/peat = 1:1). The plants were cultivated in a greenhouse after germination under 16 h of light and 8 h of darkness at 25 °C during the day and 22 °C at night (Changchun City, Jilin Province, China). The *S. alopecuroides* seedlings were cultivated in potting soil for 3 weeks. The soil was then gently rinsed off with distilled water, and the seedlings were transferred to 1/8 × Hoagland nutrient solution for 1 week. Four-week-old seedlings were treated with 1.2% NaCl while 1/8* Hoagland nutrient solution was used for the control group [4]. The treatment group included salt treatment for 4 h, 24 h, 48 h, and 72 h. At each time point, 0.5 g of *S. alopecuroides* root tissue was collected from each sample, quick-frozen in liquid nitrogen, and stored at −80 °C until use.

### 4.2. Transcriptome Sequencing Analysis and STEM Analysis of DEGs

To evaluate expression changes at the transcriptional level of *S. alopecuroides* after salt stress, we performed RNA-seq analysis on the root specimens of *S. alopecuroides* at 0 h, 4 h, 24 h, 48 h, and 72 h of salt stress. The specific experimental procedures and methods have been described previously [4]. We determined the DEGs resulting from salt stress for CK_vs_T4, CK_vs_T24, CK_vs_T48, and CK_vs_T72. The expression trends of the quantitative results of all DEGs were analyzed using the STEM analysis tool of the Lianchuan Biological Cloud Platform (https://www.omicstudio.cn/index, accessed on 4 March 2021). We selected DEGs with consistent expression trends for subsequent analysis.

### 4.3. Non-Targeted Metabolite Detection and STEM Analysis of DMs

To further reveal the influence of salt stress on the metabolism of *S. alopecuroides*, we used ultra-high-performance liquid chromatography (UHPLC) and high-resolution mass spectrometry (HRMS) to detect and quantitate metabolites. The control and salt stress *S. alopecuroides* root tissues were evaluated at 24 h, 48 h, and 72 h. Each group included six biological replicates. The specific experimental methods and procedures were previously described [4]. Differential expression analysis of all detected metabolites was performed. The DMs of CK_vs_T24, CK_vs_T48, and CK_vs_T72 were screened and change trend analysis was performed. This method is referred to as the STEM analysis method for DEGs.

### 4.4. KEGG Enrichment Analysis of DEGs

We used KEGG orthology-based annotation system (KOBAS) software to perform KEGG pathway enrichment analysis of the DEG sets. The enrichment analysis was based on the principle of hypergeometric distribution in which the DEG gene set was the DEGs identified through analysis of significant expression differences and annotated to the KEGG database (https://www.genome.jp/kegg/pathway.html, accessed on 21 March 2021) and the background gene set was the set of all the genes that underwent the significant difference analysis and annotated to the KEGG database. Pathway enrichment was used to determine the most important biochemical metabolic pathways and signal transduction pathways associated with the DEGs. To further analyze the metabolic pathways through which the DEGs of *S. alopecuroides* roots responded to salt stress, the identified DEGs were annotated to the KEGG database. The enriched pathways were classified and annotated, and the significantly enriched pathways were selected for further analysis.

### 4.5. Joint Analysis of DEGs and DMs of Phytohormone Signal Transduction Pathways

The expression of all DEGs in each group was further analyzed by annotation to the plant hormone signal transduction pathway. We analyzed the fragments per kilo base per million mapped reads (FPKM) values of the DEGs in the signal transduction pathways of AUX, CK, GA, ABA, ETH, BR, JA, and SA to determine their expression trends. We also analyzed the DEGs associated with the process of plant hormone biosynthesis. The role of each plant hormone in the response to salt stress in the roots of *S. alopecuroides* was determined by combining the changes in DM of the biosynthesis and signal transduction pathways of each plant hormone identified through the metabolome analysis.

### 4.6. Analysis of the PPI Network of the DEGs

To further analyze the relationship between the DEGs, we used the STRING PPI database (http://string-db.org/, accessed on 6 April 2021) to predict and analyze the interactions between the screened proteins. We first compared the sequences of the DEG set to the protein sequences of the reference species (soybean) in the STRING database using blastx (*e*-value = 1 × 10^−10^). The PPI relationship of the reference species was used to construct an interaction network of the DEGs.

### 4.7. qRT-PCR Quantitative Verification of DEGs and Statistical Analysis

To verify the credibility of the expression trends of the DEGs in each signaling pathway, we randomly selected 20 of the DEGs for qRT-PCR quantitative analysis. The sampling conditions were the same as those of the transcriptome sequencing sampling group. Three biological replicates and three technical replicates were analyzed. Instrument: CFX Connect Real-Time PCR Instrument (Bio-Rad, Hercules, CA, USA); reagent: 2× NovoStar^®^SYBR qPCR SuperMix Plus (Novoprotein, Lot. No: 0509941); 20 µL reaction volume; cycling program: 95 °C 2 min; 40 cycles of 95 °C 30 s, 60 °C 30 s, 72 °C 30 s; the PCR amplicon lengths were assessed using melt curve analysis. Quantitative gene primers are provided in Appendix A. The quantitative results of DEGs were analyzed for the significance of differences using the t-test analysis method. The quantitative results of each treatment and the control were statistically analyzed using the *t*-test. ** represents a significant difference of *p* < 0.01; * represents a significant difference of *p* < 0.05. IBM SPSS 19.0 statistical software (SPSS. Inc., Chicago, IL, USA) and Microsoft Excel 2010 were used for data analysis.

## Figures and Tables

**Figure 1 ijms-22-07313-f001:**
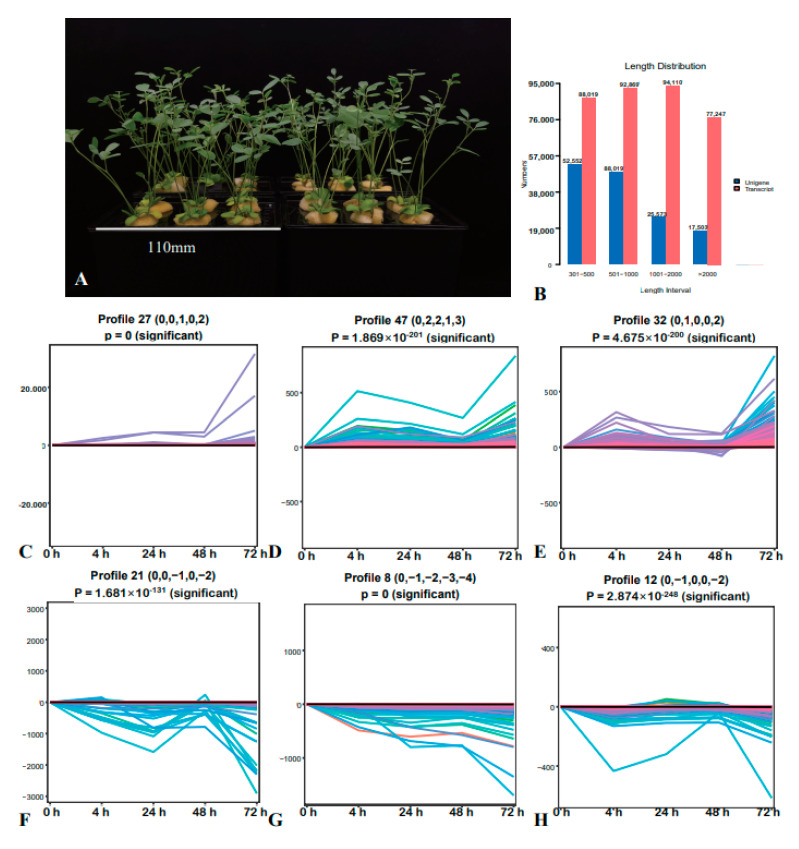
Analysis of the expression trend of differentially expressed genes. The horizontal axis represents the salt treatment time (0, 4, 24, 48, and 72 h). (**A**) Four-year-old *Sophora alopecuroides* hydroponic culture (1/8*Hoagland nutrient solution). (**B**) Transcript/unigene sequence length distribution (the abscissa is the transcript/unigene length interval, the ordinate is the number of transcripts/unigene). (**C**) Expression trend was 0.0, 0.0, 1.0, 0.0, and 2.0, with a total of 1984 differentially expressed genes (DEGs; 22.94%). (**D**) Expression trend was 0.0, 2.0, 2.0, 1.0, and 3.0, with a total of 481 DEGs (5.56%). (**E**) Expression trend was 0.0, 1.0, 0.0, 0.0, and 2.0, with a total of 1283 DEGs (14.83%). (**F**) Expression trend was 0.0, 0.0, −1.0, 0.0, and −2.0, with a total of 355 DEGs (4.10%). (**G**) Expression trend was 0.0, −1.0, −2.0, −3.0, and −4.0, with a total of 725 DEGs (8.38%). (**H**) Expression trend was 0.0, −1.0, 0.0, 0.0, and −2.0, with a total of 524 DEGs (6.06%).

**Figure 2 ijms-22-07313-f002:**
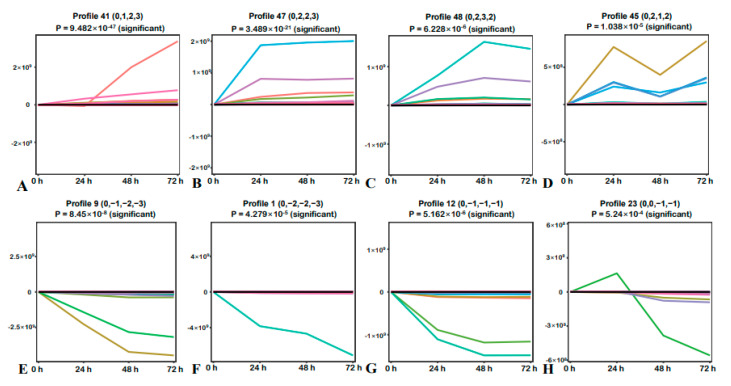
Analysis of the changing trend in metabolites. The horizontal axis represents the salt treatment time (0, 24, 48, and 72 h). (**A**) Changing trend was 0.0, 1.0, 2.0, and 3.0, with a total of 110 differentially expressed genes (DEGs; 10.01%). (**B**) Changing trend was 0.0, 2.0, 2.0, and 3.0, with a total of 72 DEGs (6.55%). (**C**) Changing trend was 0.0, 2.0, 3.0, and 2.0, with a total of 56 DEGs (5.10%). (**D**) Changing trend was 0.0, 2.0, 1.0, and 2.0, with a total of 39 DEGs (3.55%). (**E**) Changing trend was 0.0, −1.0, −2.0, and −3.0, with a total of 47 DEGs (4.28%). (**F**) Changing trend was 0.0, −2.0, −2.0, and −3.0, with a total of 42 DEGs (3.82%). (**G**) Changing trend was 0.0, −1.0, −1.0, and −1.0, with a total of 60 DEGs (5.46%). (**H**) Changing trend was 0.0, 0.0, −1.0, and −1.0, with a total of 35 DEGs (3.18%).

**Figure 3 ijms-22-07313-f003:**
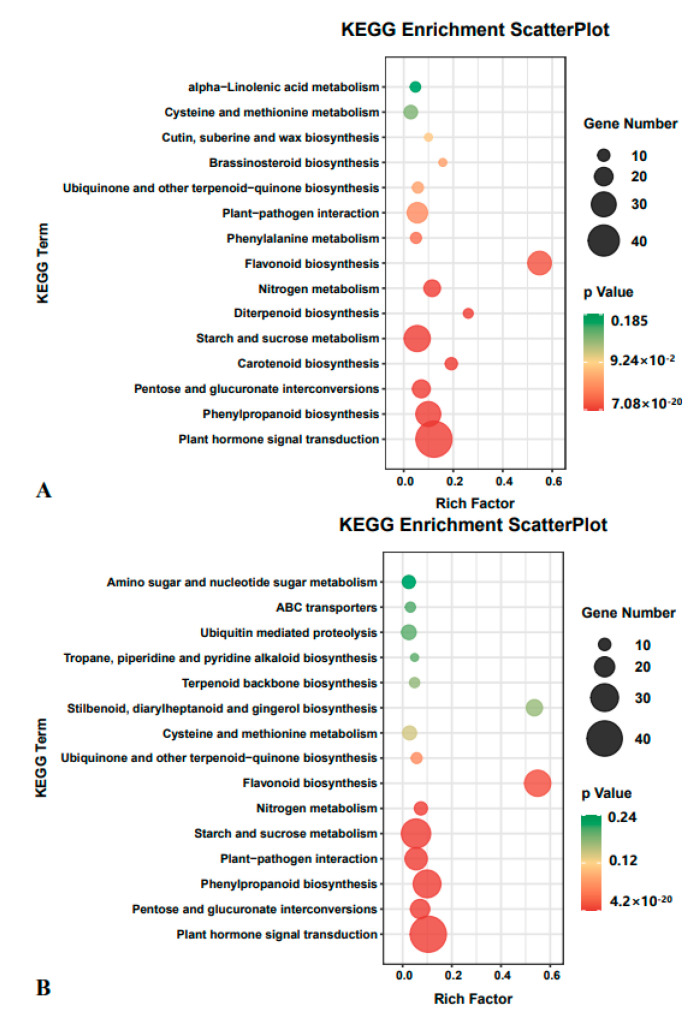
Candidate differentially expressed gene Kyoto Encyclopedia of Genes and Genomes (KEGG) enrichment factor map, sorting the top 15 pathways according to the *p*-value. (**A**) Upregulated expression trend in differential gene enrichment results; (**B**) downregulated expression trend in differential gene enrichment results.

**Figure 4 ijms-22-07313-f004:**
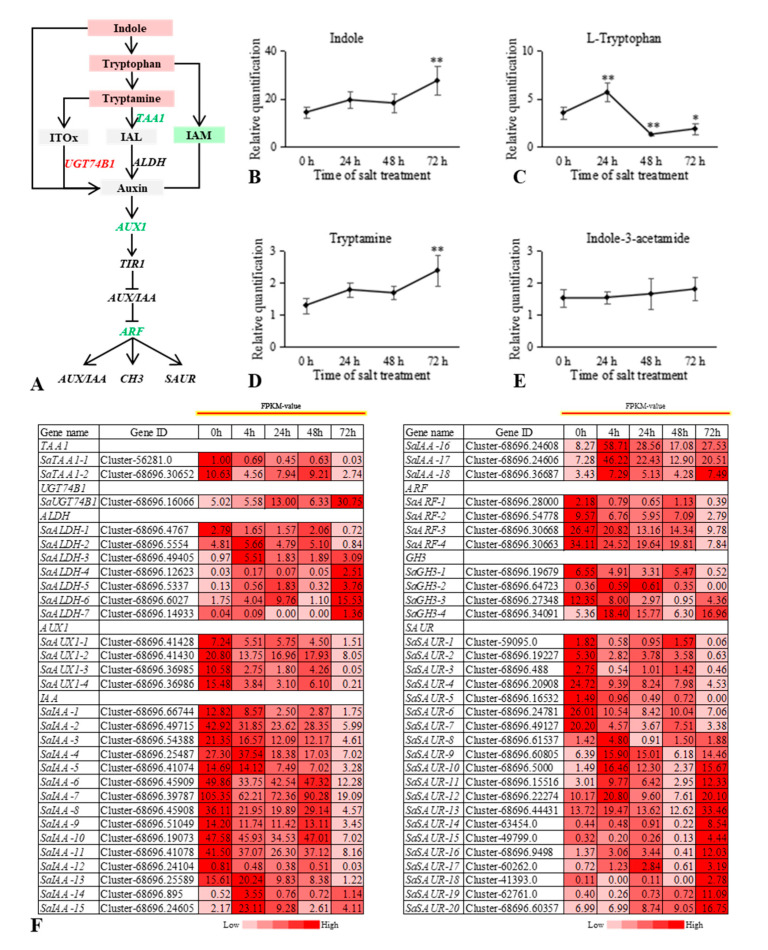
Overview of the relationship between differentially expressed genes (DEGs) and differential metabolites (DMs) in the auxin signaling pathway of *Sophora alopecuroides* under salt stress. (**A**) Overview of auxin signaling pathway. (**B**–**E**) The trend in the auxin signaling pathway DM changes with salt stress. Expression scores are shown as fold change. The horizontal axis represents the duration of salt treatment, and the vertical axis represents the relative quantification of metabolites (peak area × 10^6^). Expression levels of six independent samples of metabolites and the control were compared by *t*-test, where ** represents *p* < 0.01 and * represents *p* < 0.05. (**F**) Heat map of auxin signaling pathway-related gene expression. Values are average fragments per kilo base per million mapped reads (FPKM) values of each sample in each group. ALDH, aldehyde dehydrogenase (NAD+); ARF, auxin response factor; AUX1, auxin influx carrier (AUX1 LAX family); IAA, auxin-responsive protein IAA; IAL, indole-3-acetaldehyde; IAM, indole-3-acetamide; ITOx, indole-3-thiohydroximate; SAUR, SAUR family protein; GH3, auxin responsive GH3 gene family.

**Figure 5 ijms-22-07313-f005:**
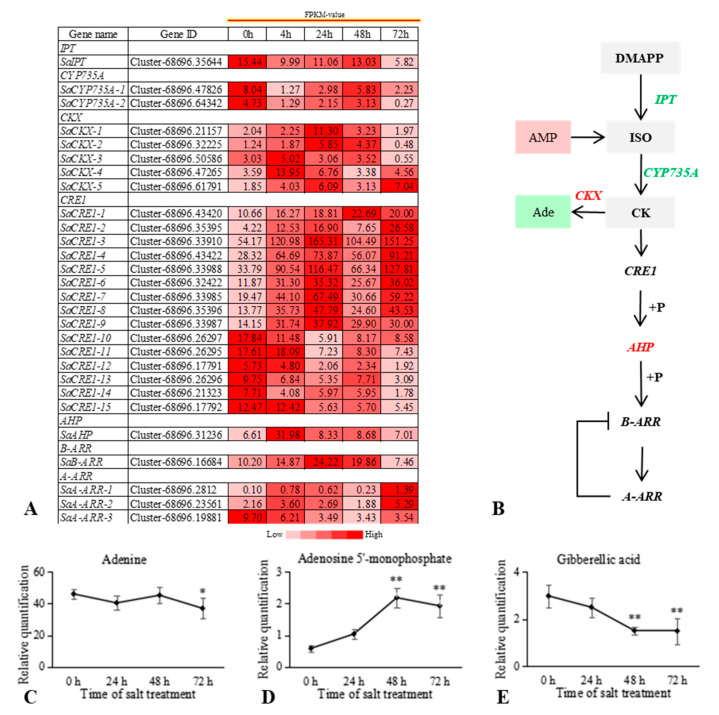
Overview of the relationship between differentially expressed genes (DEGs) and differential metabolites (DMs) in the CK signaling pathway of *S. alopecuroides* under salt stress. (**A**) Heat map of CK signaling pathway-related gene expression. Values are average FPKM values of each sample in each group. (**B**) Overview of CK signaling pathway. (**C**–**E**) The trend in CK and gibberellin (GA) signaling pathway DM changes with salt stress. The horizontal axis represents the duration of salt treatment, and the vertical axis represents the relative quantification of metabolites (peak area × 10^6^). Expression levels of six independent samples of metabolites and the control were compared by *t*-test, where ** represents *p* < 0.01 and * represents *p* < 0.05. Expression scores are shown as fold-change. Ade, adenine; CRE1, *Arabidopsis* histidine kinase 2/3/4 (cytokinin receptor); AHP, histidine-containing phosphotransfer protein; ARR-A, two-component response regulator ARR-A family; ARR-B, two-component response regulator ARR-B family; CKs, cytokinins; CKX, cytokinin dehydrogenase; CYP735A, cytokinin trans-hydroxylase; ISO, isopentenyl; IPT, cystathionine beta-lyase.

**Figure 6 ijms-22-07313-f006:**
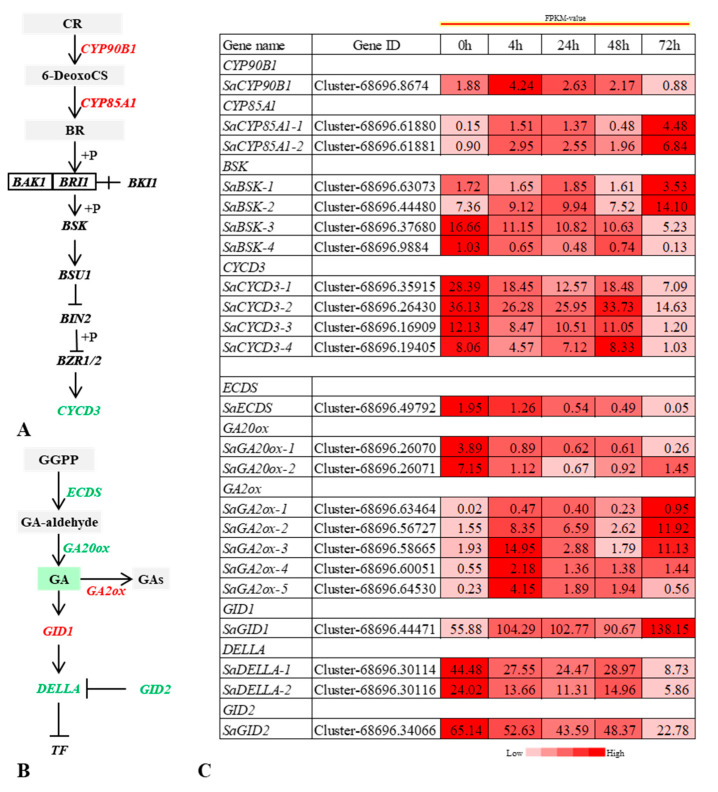
Overview of the relationship between differentially expressed genes (DEGs) and differential metabolites (DMs) in the BR and gibberellin (GA) signaling pathway of *Sophora alopecuroides* under salt stress. (**A**) Overview of BR signaling pathway. (**B**) Overview of GA signaling pathway. (**C**) Heat map of CK and GA signaling pathway-related gene expression. Values are average FPKM value of each sample in each group. BRs, brassinosteroids; BSK, BR-signaling kinase; CR, campesterol; CYCD3, cyclin D3; CKs, cytokinins; DELLA, DELLA protein; GA20ox, gibberellin 20-oxidase; GA2ox, gibberellin 2-oxidase; GGPP, geranglgeranyl-PP; GID1, gibberellin receptor GID1; GID2, F-box protein GID2.

**Figure 7 ijms-22-07313-f007:**
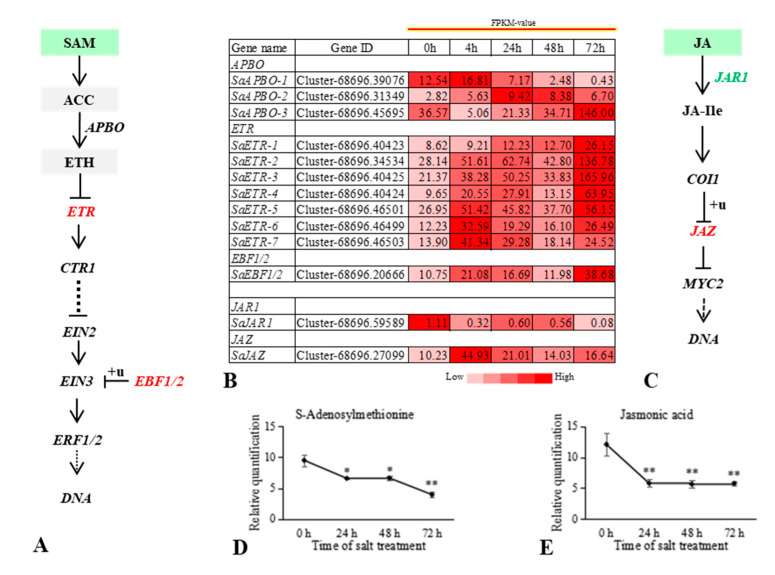
Overview of the relationship between differentially expressed genes (DEGs) and differential metabolites (DMs) in the ETH and JA signaling pathway of *Sophora alopecuroides* under salt stress. (**A**) Overview of ETH signaling pathway. (**B**) Heat map of ETH and JA signaling pathway-related gene expression. Values are average FPKM value of each sample in each group. (**C**) Overview of JA signaling pathway. (**D**,**E**) The trend in ETH and JA signaling pathway DM changes with salt stress. The horizontal axis represents the duration of salt treatment, and the vertical axis represents the relative quantification of metabolites (peak area × 10^6^). Expression levels of six independent samples of metabolites and the control were compared by *t*-test, where ** represents *p* < 0.01 and * represents *p* < 0.05. Expression scores are shown as fold-change. ACC, 1-aminocyclopropane-1-carboxylate; APBO, aminocyclopropanecarboxylate oxidase; EBF1/2, EIN3-binding F-box protein; ERF1, ethylene-responsive transcription factor 1; ETH, ethylene; ETR, ethylene receptor; JA, jasmonate; JAR1, jasmonic acid-amino synthetase; JAZ, jasmonate ZIM domain-containing protein; SAM, *S*-adenosyl-*l*-methionine.

**Figure 8 ijms-22-07313-f008:**
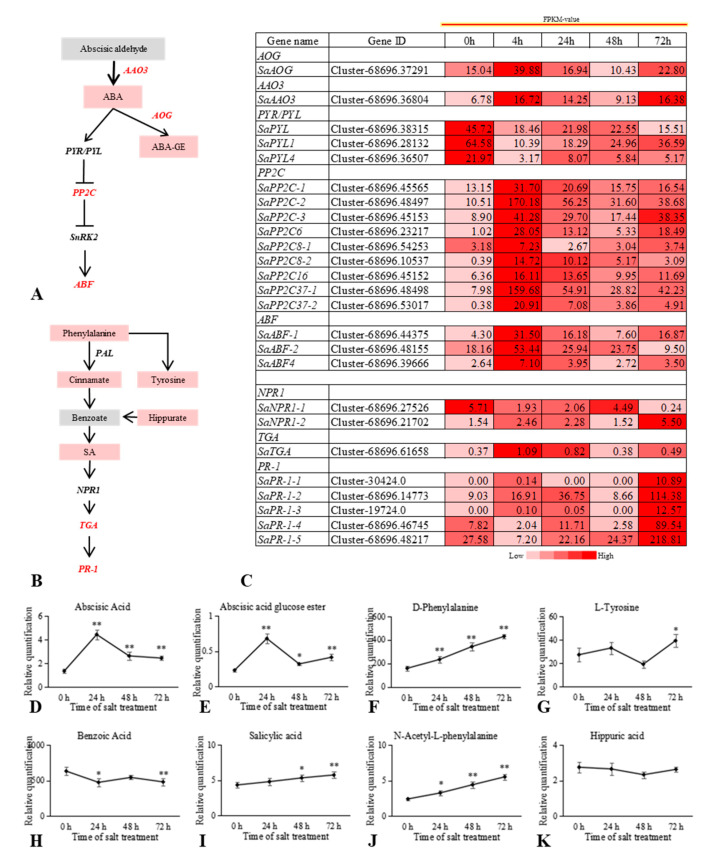
Overview of the relationship between differentially expressed genes (DEGs) and differential metabolites (DMs) in the ABA and SA signaling pathway of *Sophora alopecuroides* under salt stress. (**A**) Overview of ABA signaling pathway. (**B**) Overview of SA signaling pathway. (**C**) Heat map of ABA and SA signaling pathway-related gene expression. Values are average FPKM value of each sample in each group. (**D**–**K**) The trend in ABA and SA signaling pathway DM changes with salt stress. The horizontal axis represents the time of salt treatment, and the vertical axis represents the relative quantification of metabolites (peak area × 10^6^). Expression levels of six independent samples of metabolites and the control were compared by *t*-test, where ** represents *p* < 0.01 and * represents *p* < 0.05. Expression scores are shown as fold-change. AAO3, abscisic-aldehyde oxidase; ABA, abscisic acid; ABA-GE, abscisic acid glucose ester; ABF, ABA responsive element binding factor; NPR1, regulatory protein NPR1; PAL, phenylalanine ammonia-lyase; PP2C, protein phosphatase 2C; PR-1, pathogenesis-related protein 1; PYL, abscisic acid receptor PYR/PYL family; SA, salicylate; SnRK2, *Abrus precatorius* serine/threonine-protein kinase SAPK2-like; TGA, transcription factor TGA.

**Figure 9 ijms-22-07313-f009:**
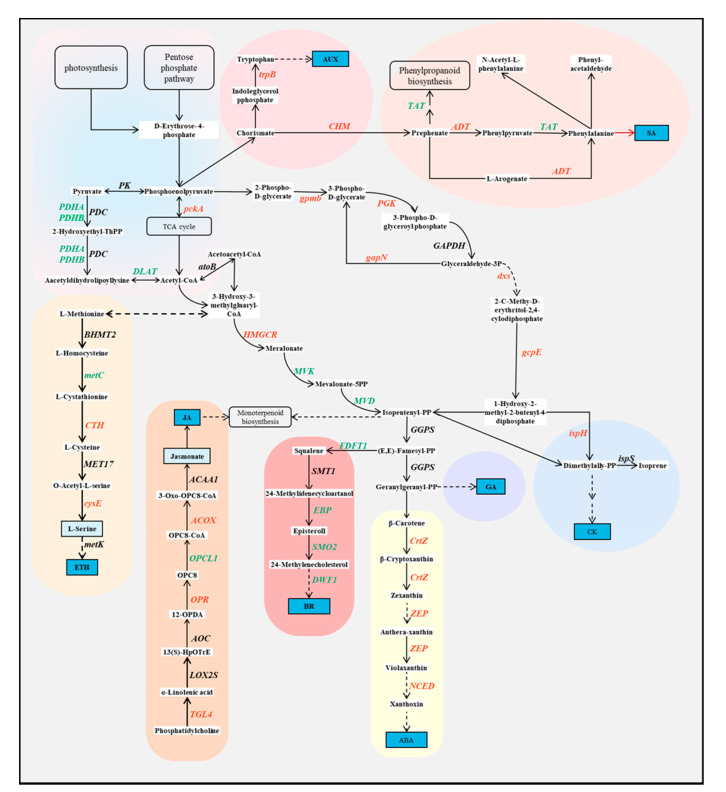
Overview of phytohormone biosynthetic pathway. Red represents the upregulated expression of the corresponding DEGs under salt stress, green represents the downregulated expression of DEGs, and black represents the presence of both upregulated and downregulated DEGs. The corresponding expression level of each DEG is shown in Appendix A. AUX, auxin; trpB, tryptophan synthase beta chain; 4CL, 4-coumarate-CoA ligase; ACAA1, acetyl-CoA acyltransferase 1; ACOX1, acyl-CoA oxidase; ADT, arogenate/prephenate dehydratase; ADT, PDT arogenate/prephenate dehydratase; AOC, allene oxide cyclase; AOG, abscisate beta-glucosyltransferase; atoB, acetyl-CoA C-acetyltransferase; AUX, auxin; BHMT2, homocysteine *S*-methyltransferase; CDK2, cyclin-dependent kinase 2; CHI, chalcone isomerase; CHM, chorismate mutase; CHS, chalcone synthase; CKS1, cyclin-dependent kinase regulatory subunit CKS1; crtZ, beta-carotene 3-hydroxylase; CTH, cystathionine gamma-lyase; CYP73A, trans-cinnamate 4-monooxygenase; CYP85A1, brassinosteroid-6-oxidase 1; CYP90B1, steroid 22-alpha-hydroxylase; CYP93C, 2-hydroxyisoflavanone synthase; cysE, serine *O*-acetyltransferase; cysK, cysteine synthase A; DLAT, pyruvate dehydrogenase E2 component (dihydrolipoamide acetyltransferase); DLD, dihydrolipoamide dehydrogenase; DMAPP, dimethylally-PP; DWF1, delta24-sterol reductase; dxs, 1-deoxy-d-xylulose-5-phosphate synthase; EBP, cholestenol delta-isomerase; ECDS, ent-copalyl diphosphate synthase; FDFT1, farnesyl-diphosphate farnesyltransferase; GAPDH, glyceraldehyde 3-phosphate dehydrogenase; gapN, glyceraldehyde-3-phosphate dehydrogenase (NADP+); gcpE, (E)-4-hydroxy-3-methylbut-2-enyl-diphosphate synthase; GGPS, geranylgeranyl diphosphate synthase, type II; gpmB, probable phosphoglycerate mutase; HMGCR, hydroxymethylglutaryl-CoA reductase (NADPH); ispH, 4-hydroxy-3-methylbut-2-en-1-yl diphosphate reductase; ispS, isoprene synthase; LED, leucoanthocyanidin dioxygenase; LOX2S, lipoxygenase; metK, S-adenosylmethionine synthetase; MPK6, mitogen-activated protein kinase 6; MVD, diphosphomevalonate decarboxylase; MVK, mevalonate kinase; NCED, 9-cis-epoxycarotenoid dioxygenase; OPCL1, OPC-8:0 CoA ligase 1; OPR, 12-oxophytodienoic acid reductase; pckA, phosphoenolpyruvate carboxykinase (ATP); PDC, pyruvate decarboxylase; PDHA, pyruvate dehydrogenase E1 component alpha subunit; PDHB, pyruvate dehydrogenase E1 component beta subunit; PGK, phosphoglycerate kinase; PK, pyruvate kinase; SMO2, 4-alpha-methyl-delta7-sterol-4alpha-methyl oxidase; SMT1, sterol 24-C-methyltransferase; TAA1, l-tryptophan-pyruvate aminotransferase; TAT, tyrosine aminotransferase; TGL4, TAG lipase/steryl ester hydrolase/phospholipase A2/LPA acyltransferase; trpB, tryptophan synthase beta chain; UGT74B1, *N*-hydroxythioamide *S*-beta-glucosyltransferase; ZEP, zeaxanthin epoxidase.

**Figure 10 ijms-22-07313-f010:**
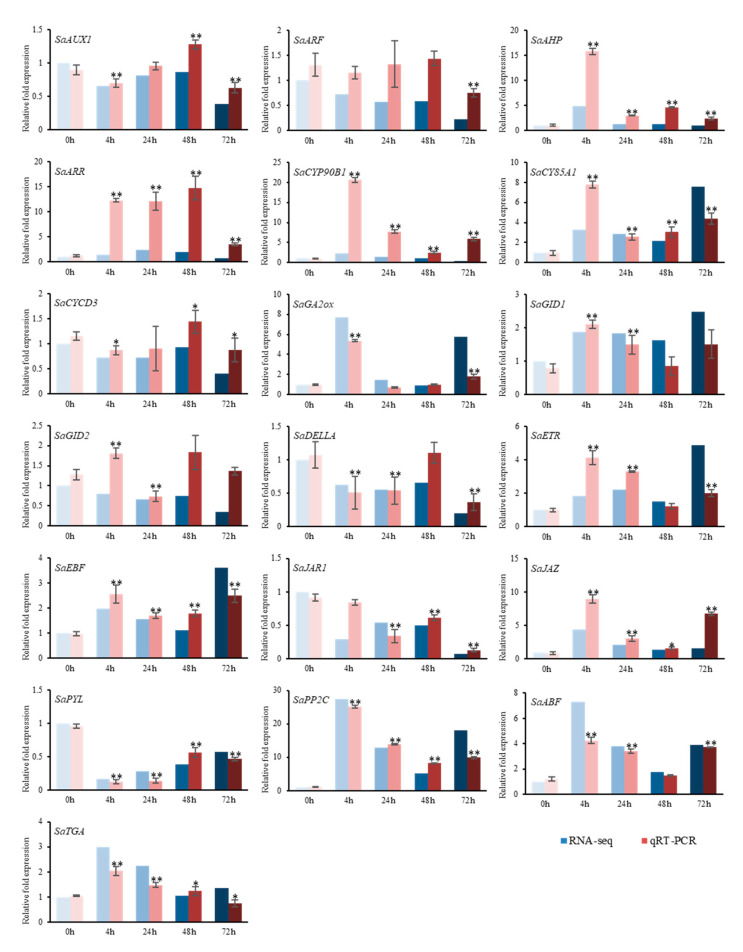
qRT-PCR verification of differentially expressed genes (DEGs). The relative gene expression levels under 1.2% NaCl treatment at different periods. Vertical bar indicates the mean ± SD calculated from three replicates. Statistical comparisons (one-way analysis of variance (ANOVA)) are presented for each variable (** *p* < 0.01 * *p* < 0.05).

## Data Availability

Not applicable.

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
