# Peer review of "Analysis of Phytohormone Signal Transduction in Sophora alopecuroides under Salt Stress"

_ijms, 2021, doi:10.3390/ijms22147313_

Round 1

Reviewer 1 Report

In continuation to their work on the non-model plant species Sophora alopecuroides, the Authors proposed an RNAseq transcriptomic analysis coupled with metabolomics to decipher the role of phytohormones in the response to salinity. Salinity is an important emerging issue for crops including non-model species, this work is therefore of special interest in this context.

The paper is well writing (I have only very minor typos corrections to suggest). The experiments are well performed and their description in Materials and Methods sufficient. The figures and tables are well prepared and comprehensive.

This work is in the scope of IJMS and in the range of the published papers in terms of quality.

Therefore, I consider that this work deserve publication in IJMS.

Only minor typos mistakes have to be corrected: such as small capital uses for “L” or “D” for example for amino acids, “O” or “N” in italics when indicating substitution of O or N atoms, genes names in italics and on the contrary protein names are not, …

Author Response

Thank you for your suggestion. We have checked and revised the full text. The specific revisions are as follows:

Line 302: S-adenosyl-l-methionine

Line 336: l-phenylalanine, d-phenylalanine, and N-acetyl-l-phenylalanine.

Line 380: homocysteine S-methyltransferase

Line 384: serine O-acetyltransferase

Line 387: 1-deoxy-d-xylulose-5-phosphate synthase

Line 393: 4-hydroxy-3-methylbut-2-en-1-yl diphosphate reductase

Line 402: l-tryptophan-pyruvate aminotransferase

Line 404: N-hydroxythioamide S-beta-glucosyltransferase

Reviewer 2 Report

In the title, the Authors indicate that they analyzed the role of the phytohormone signal transduction pathway in response to salt stress in Sophora alopecuroides. But in my opinion, the Authors indicate only dependencies between salt stress and transcriptomic or metabolomic changes. To showing the role of plant hormones' signal transduction pathway, the Authors should study the mutants with knock-out selective of genes. In the present form, there are only cause-effect relationships. 

Moreover, the studies were conducted on roots, but the reader learns about it in the Material and methods section.

In part 2.1 the Authors wrote about the hydroponic condition, but in Material and methods, I can see that plants were sown in potting soil (vermiculite:peat=1:1). It is not clear. In Fig.1A we can see only shoots parts. In my opinion, it would be better to show how salt stress affects root growth.

In Fig. 4 b-e, 5 c-e, 7 d-e, 8 b-k, there is a lack of standard deviation, and it hasn't been know if the changes are statistically significant.

I suggest the Fig.9  move to Introduction.

In the Supplementary files, there is no Video S1.

The Authors should clearly indicate that the experiments were done on roots.

I strongly suggest rewriting and reorganize this manuscript.

Author Response

Thank you for your suggestion.

Round 2

Reviewer 2 Report

Thank you very much for your comprehensive comments on my review of your manuscript. The text has been amended following my comments and requirements and was significantly improved.